# Shifts in the selectivity filter dynamics cause modal gating in K$^+$ channels

Shehrazade Jekhmane[1], João Medeiros-Silva [1], Jing Li [2], Felix Kümmerer [1], Christoph Müller-Hermes[1], Marc Baldus [1], Benoît Roux[2] & Markus Weingarth [1]

Spontaneous activity shifts at constant experimental conditions represent a widespread regulatory mechanism in ion channels. The molecular origins of these modal gating shifts are poorly understood. In the K$^+$ channel KcsA, a multitude of fast activity shifts that emulate the native modal gating behaviour can be triggered by point-mutations in the hydrogen bonding network that controls the selectivity filter. Using solid-state NMR and molecular dynamics simulations in a variety of KcsA mutants, here we show that modal gating shifts in K$^+$ channels are associated with important changes in the channel dynamics that strongly perturb the selectivity filter equilibrium conformation. Furthermore, our study reveals a drastically different motional and conformational selectivity filter landscape in a mutant that mimics voltage-gated K$^+$ channels, which provides a foundation for an improved understanding of eukaryotic K$^+$ channels. Altogether, our results provide a high-resolution perspective on some of the complex functional behaviour of K$^+$ channels.

[1] NMR Spectroscopy, Bijvoet Center for Biomolecular Research, Department of Chemistry, Faculty of Science, Utrecht University, Padualaan 8, 3584 CH Utrecht, The Netherlands. [2] Department of Biochemistry and Molecular Biology, The University of Chicago, 929 E57th Street, Chicago, IL 60637, USA. These authors contributed equally: Shehrazade Jekhmane, João Medeiros-Silva. Correspondence and requests for materials should be addressed to M.W. (email: m.h.weingarth@uu.nl)

Potassium (K[+]) channels are of fundamental importance for the functioning of excitable cells[1]. They allow selective and rapid flux of K[+] across the cell membrane through a central pore, which is regulated by the interplay between a cytoplasmic activation gate and an extracellular C-type inactivation gate known as selectivity filter. The selectivity filter sequence TVGYG is highly conserved, and its backbone carbonyl-groups together with the threonine hydroxyl group line up to form the five K[+] coordination sites (S0–S4)[2,3].

Extensive crystallographic studies in the well-accepted model K[+] channel KcsA showed that C-type inactivation is governed by a complex hydrogen bond network behind the selectivity filter[4,5]. Residue E71 is at the centre of this network, and modulates the selectivity filter by coordinating to the backbone of Y78 and, mediated via a water molecule, the D80 as well as the W67 side chains (Fig. 1a). While W67 and D80 are highly conserved in K[+] channels, E71 is commonly replaced by a valine or isoleucine in eukaryotes (Fig. 1b), which is assumed to critically modulate selectivity filter gating. Indeed, electrophysiological measurements showed that point-mutations at E71 lock the KcsA channel into different, natively occurring gating modes, which are best represented by a high-open probability (E71A), a low-open probability (E71I), and a high-frequency flicker (E71Q) mode[6]. Random shifts between such gating modes, known as modal gating shifts, were observed in various eukaryotic and prokaryotic K[+] channels, and are a widespread regulatory mechanism of channel activity[4,7–11]. Yet, despite their broad functional importance, the structural correlates and triggers of modal gating shifts are unknown. Modal gating shifts were suggested to relate to selectivity filter rearrangements; however, a series of X-ray structures of E71X mutants showed no changes in the filter (RMSD relative to WT KcsA is <0.25 Å)[6] despite the marked functional heterogeneity of these mutants. Curiously, for the E71A mutant, a well-established model to study K[+] channel gating[12,13], a second, strongly different filter conformation of uncertain functional relevance was crystallised[4,14]. Besides the lack of clarity on the selectivity filter conformation, it was further assumed that changes in the filter dynamics could cause modal gating shifts[6]. However, also here, whether the selectivity filter dynamics change in reference to the gating mode is unknown, and experimental data are scarce to resolve this question. Altogether, there is a fundamental lack of knowledge on how the hydrogen bond network surrounding the selectivity filter modulates its gating, which critically limits our understanding of modal gating shifts of Kv channels.

Here, we use modern proton-detected (1H-detected)[15–20] solid-state NMR (ssNMR) in native-like membranes to compare the selectivity filter in WT KcsA and the three mutants (E71A, E71I, E71Q) that are best representatives of modal gating. We show that E71 point-mutations cause marked changes in the selectivity filter conformational dynamics, in contrast to previous crystallographic studies that revealed virtually no changes in structure[6]. By combining ssNMR with molecular dynamics (MD) simulations, we demonstrate that altered structural dynamics in E71X mutants drive the selectivity filter into new conformational equilibria that represent the molecular origin of modal gating. Furthermore, we show that modal gating goes hand in hand with fluctuations in the hydrogen bonding and water network behind the filter, which are triggers of sudden mode shifts. Altogether, these results provide a high-resolution perspective on the complex kinetic behaviour of the selectivity filter of K[+] channels. Importantly, the pronounced conformational and motional changes that we observe in E71I KcsA provide a foundation for future elucidation of the selectivity filter of eukaryotic K[+] channels.

## Results

**NMR assignments of the channels at near-native conditions**. We assigned the 1H, 13C, and 15N ssNMR chemical shifts of WT KcsA and the E71A, E71I, E71Q mutants in order to analyse their conformational dynamics in membranes. First, we prepared uniformly [13C,15N]-labelled inversely Fractionally Deuterated[17] channels in liposomes composed of *Escherichia coli* lipids. Samples were prepared in buffer conditions (pH 7.4, 100 mM K[+]) at which WT KcsA is in the closed-conductive state[21], i.e., a state with a closed activation gate and a conductive selectivity filter. De novo backbone chemical shift assignments were obtained for mutant E71A using a set of four dipolar-based three-dimensional (3D) 1H-detected ssNMR experiments (CANH, CONH, CAcoNH, COcaNH) (Fig. 1c and Supplementary Figure 1). The high spectral quality enabled us to almost fully assign residues L41–W87, which include the complete selectivity filter and pore helix, the pore loop, larger parts of the outer transmembrane 1 (TM1) helix, and a few residues of the inner transmembrane 2 (TM2) helix. Since we used dipolar-based magnetisation transfer steps that decrease in efficiency with increasing molecular mobility, the cytoplasmic domain (F125-R160) and the membrane-associated M0 helix (M1-H20) were not detectable at our experimental temperatures of 300–310 K[22]. Assignments were then transferred to KcsA mutants E71I and E71Q and confirmed with a reduced set of 3D ssNMR experiments (CANH, CONH). The reduced set was also used to complement our previous WT KcsA assignments[17]. For the flicker mutant E71Q, spectral sensitivity was significantly lower, indicative of increased mobility, which averages dipolar magnetisation transfer efficiency. Furthermore, for all mutants and WT KcsA, we acquired two-dimensional (2D) 13C–13C experiments. Moreover, we acquired dipolar 2D 15N–1H ssNMR spectra, in which each signal relates to one 15N–1H backbone or side chain unit, and which represent spectral fingerprints.

**Modal gating relates to changes in the filter conformation**. NMR chemical shifts are sensitive reporters of protein conformation. Therefore, comparing the chemical shifts of the mutants to WT KcsA enables the analysis of the structural impact of the substitutions at E71. The 2D 15N–1H and 13C–13C spectra of all E71X mutants superimposed very well onto WT KcsA, demonstrating that the global protein fold is conserved (Supplementary Figure 2). However, for all mutants, we observed remarkably large 1H and 15N (HN) as well as 13C chemical shift perturbations (CSPs) across the selectivity filter (Fig. 1d, e and Supplementary Figure 3). For all mutants, we observed very important HN CSPs for the essential residues W67, Y78, and D80 behind the filter, demonstrating changes in their hydrogen bond interactions. At the same time, the large Cα and CO CSPs that we observed for residues G77, Y78, and D80 imply pronounced conformational filter backbone changes, which likely have implications for the K[+] occupancy. Intriguingly, all mutants also exhibited a large CSP for V76CO, which is immediately involved in K[+] binding and key residue for C-type inactivation (Fig. 1e)[3].

Mutants E71A and E71I showed a strikingly similar CSP pattern. Here, the HN and 13C signal shifts of W67, V76, Y78, and D80 are all similar in magnitude and in the same direction. This similarity appears plausible, given that neither alanine nor isoleucine is able to mimic the hydrogen bonding capacities of the E71 carboxyl group. However, although the CSP pattern is similar, E71I shows a much larger Y78Cα CSP than E71A, which implies conformational differences.

While we also observed CSP maxima at W67, G77, Y78, and D80 in E71Q, these signal shifts were partially in opposite direction relative to E71A and E71I, and less strong. In general,

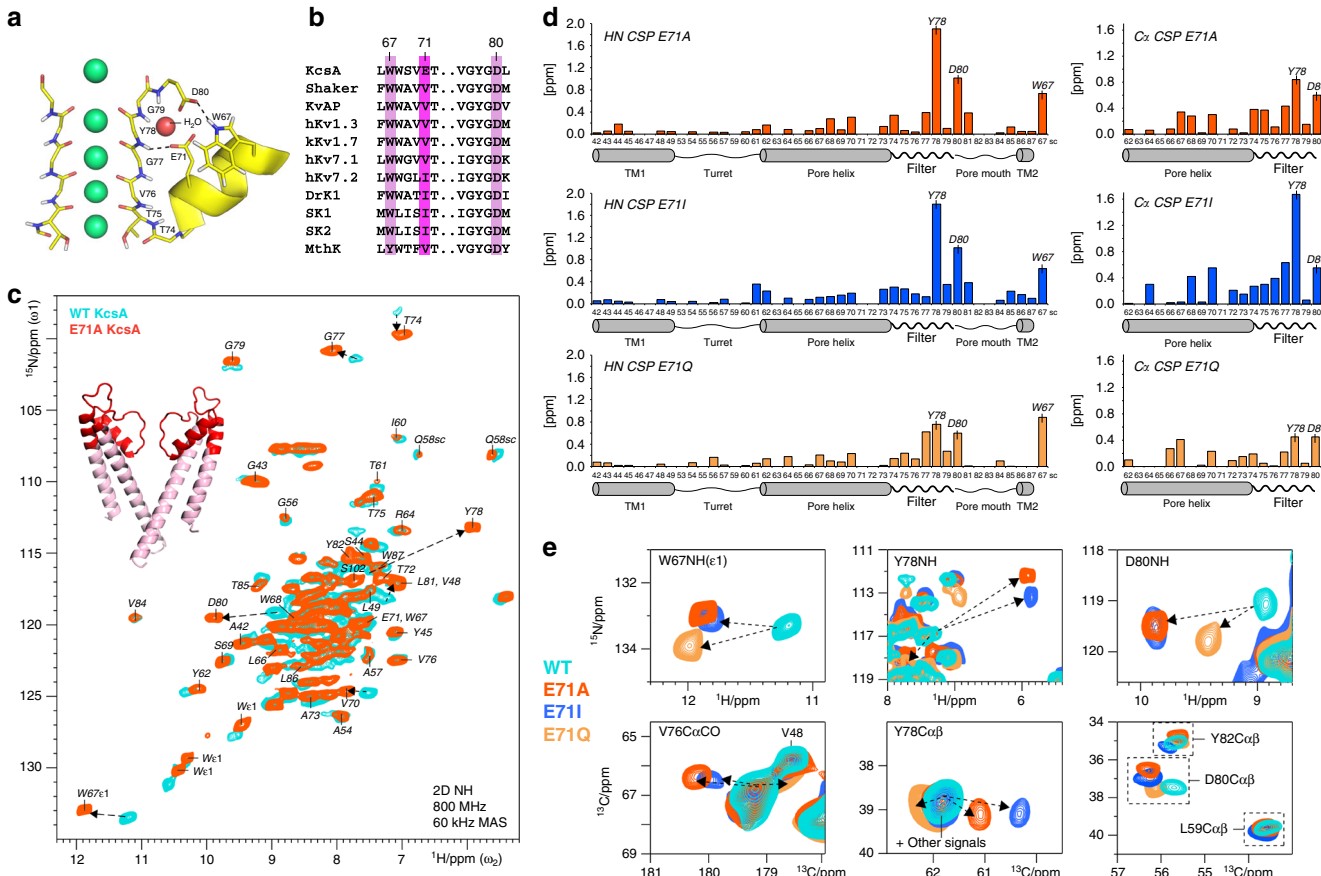

**Fig. 1** E71X point-mutations cause large conformational changes in the KcsA selectivity filter. **a** The selectivity filter of K$^+$ channel KcsA (1K4C) is regulated by a hydrogen bond network with the triad W67–E71–D80 at the centre[5]. **b** W67 and D80 are highly conserved, while E71 is commonly replaced by a nonpolar valine or isoleucine in Kv channels. **c** 2D NH ssNMR spectra of WT KcsA (cyan) and mutant E71A (red) acquired in membranes. Arrows indicate major signal shifts of key residues. Residues L41–W87 are annotated in the E71A spectrum and highlighted in red on the X-ray structure. **d** Chemical shift perturbations (CSPs) of E71A (red), E71I (blue), and E71Q (orange) in reference to WT KcsA. Combined HN CSPs (left) of amino-protons and backbone-nitrogens and (right) Cα CSPs. The strongest NH CSPs in E71A are highlighted in **c**. Source data are provided as a Source Data file. **e** 2D NH (upper panel) and 2D CC spectra (lower) showing large CSPs of key residues W67, V76, Y78, and D80 in E71A (red), E71I (blue), and E71Q (orange) relative to WT KcsA (cyan)

among the three mutants, the chemical shifts of E71Q deviated the least from WT KcsA. Considering that a glutamine can partly substitute for some of the hydrogen bonds of a glutamate, it is likely that the E71Q filter conformation is relatively close to WT KcsA. Indeed, the D80 side chain and the Y78 backbone showed by far the smallest CSPs for E71Q, which strongly suggests that Q71–D80 and Q71–Y78 maintain interactions analogous to WT KcsA, while these interactions are lost in E71A and E71I (Figs. 2a and 1e).

Next to conformational changes in the filter, the mutants showed moderate to larger CSPs around residues W67–V70. These changes presumably relate to modulations in the aromatic belt W67, W68, and Y78 that surrounds the filter, and to the loss or modulation of interaction E71–D80, which acts as a molecular spring that couples the pore helix to the pore mouth in WT KcsA (Fig. 2b)[5]. Our data show that the pore helices in all mutant channels exhibit, compared to WT KcsA, local changes (Fig. 2d) that modulate the residues T74–T75–V76–G77 at the N-terminal end of the selectivity filter, which are directly coordinated to the pore helix by hydrogen bonds. Indeed, all mutants show clear perturbations at V76 and G77 (Fig. 1d, e). We observed the largest CSPs in the E71I pore helix, which correlates with strong signal shifts of the T74 side chain. Since conformational changes of the T74 side chain relate to C-type inactivation[23,24], this

perturbation could be functionally important for E71I, which favours transitions to a non-conductive state[6]. This assumption is corroborated by 2D CC spectra, which show that the T74 side chain conformation in E71I (pH 7.4, 100 mM K$^+$) and in the inactivated filter of WT KcsA (pH 4, 0 mM K$^+$) are close to each other (Fig. 2e, f).

Furthermore, we observed remarkable conformational changes in the turret G53–T61. The turret is an important drug binding site[25] and responds to gating changes[26]; however, the structural underpinning is unclear and inaccessible from X-ray structures because of intense interactions between turret and Fab fragments[4]. In our ssNMR experiments, the CSPs in the turret are small. However, to our surprise, residues P55, G56, and A57 in E71I disappeared or showed signal splittings, strongly indicative of stark structural heterogeneity (Fig. 2c). This implies that the replacement of E71 by isoleucine causes long-range effect that are felt ~2 nm away from the mutation site. Note that CSPs for the TM1 and TM2 helices, as well as for un-annotated TM residues, were very small, confirming that the global WT KcsA fold is conserved in the mutants.

**Changes in the filter dynamics and modal gating shifts.** Historically, the selectivity filter was thought to form a stiff

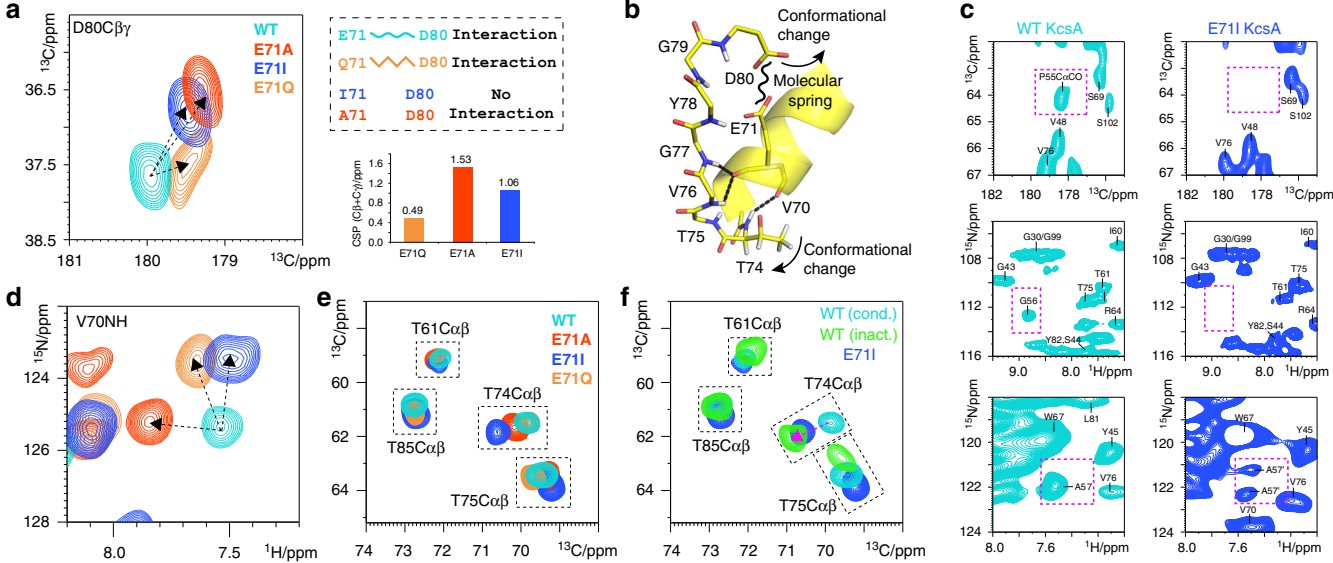

**Fig. 2** Loss of the E71–D80 interaction causes extended rearrangements behind the filter. **a** Zoom into (left) 2D CC ssNMR spectra of WT KcsA (cyan), E71A (red), E71I (blue), and E71Q (orange) showing the D80 side chain CSPs. E71Q (right) mimics the E71–D80 interaction, which is lost in E71A and E71I. The D80 side chain CSPs are large in E71A and E71I, while they are small in E71Q. **b** Structural representation (1K4C) of the stabilisation of filter residues T74–G77 by hydrogen bonds with V70 and E71 of the pore helix. **c** The E71I turret is disordered, which causes signals to disappear or split. Spectral zooms are show for WT KcsA (cyan) and E71I (blue). **d** Overlay of 2D NH spectra showing a strong CSP for V70 of the pore helix. **e** Overlay of 2D CC spectra, showing CSPs of the functionally critical T74 side chain in E71X mutants. **f** The large T74 CSP in E71I (pH 7.4, 100 mM K$^+$) is reminiscent of the inactivated filter in WT KcsA (pH 4, 0 mM K$^+$)

framework in order to allow fast conduction of K$^+$ together with high selectivity over Na$^{+2}$. More recent studies point to a more dynamic filter[27], and hypothesise that modal gating shifts relate to changes in the motional behaviour of the filter[6]. However, quantitative experimental data on selectivity filter dynamics are not available in membranes, critically limiting our understanding of K$^+$ channel function. Here we probe the filter dynamics in reference to the gating mode with ssNMR relaxation, which is an ideal approach to measure the internal dynamics of membrane proteins at native conditions[17,28]. Site-resolved ssNMR relaxation can be probed with 2D $^{15}$N–$^1$H experiments that include a relaxation element that is sensitive to dynamics on a certain time-scale. A series of spectra is then acquired with increasing duration of the relaxation element, and the signal sensitivity decreases according to the motion for a given residue. The signal decay is then converted into a relaxation rate $R$, and higher rates indicate enhanced dynamics. We probed the $^{15}$N slow rotating-frame relaxation ($R_{1rho}$) for WT KcsA and the mutants (Fig. 3) using extensive series of $^1$H-detected 2D $^{15}$N–$^1$H experiments[29]. $^{15}$N $R_{1rho}$ relaxation is sensitive to dynamics in the nanosecond–millisecond range but dominated by motion with slow correlation times in the microsecond range[30]. The high sensitivity with $^1$H-detection enabled us to measure relaxation rates with high accuracy. In all investigated channels, our study unravelled strikingly different filter dynamics that clearly corre-late with the CSP maxima, thereby linking conformational and motional changes. In WT KcsA, the filter (~15 ms$^{-1}$ $R_{1rho}$) is the most dynamic membrane-embedded region, with G79 as a dis-tinct local maximum. The dynamics of the pore helix is slightly lower, TM1 residues show the least dynamics, while the extra-cellular turret is by far the most mobile region. These results agree with our previous relaxation studies in WT KcsA[17].

The global dynamics of mutant E71I, which mimics certain Kv channels, is particularly interesting. Surprisingly, compared to WT KcsA, we measured strongly enhanced dynamics at the filter entrance Y78-D80 (~25 ms$^{-1}$ $R_{1rho}$). These residues also showed

the largest CSPs, demonstrating again that local conformational and motional changes correlate. Furthermore, we observed sizeable stiffening at G77 in the middle of the E71I filter. Intriguingly, the selectivity filter dynamics is very different in the E71A mutant, in which the middle and upper filter regions (V76–D80) drastically rigidified (~8 ms$^{-1}$ $R_{1rho}$). This means that E71A and E71I exhibit clearly different filter dynamics in spite of similar CSP patterns. It is easy to imagine that these differential dynamics are at the origin of some of the heterogeneous gating kinetics observed in KcsA and other K$^+$ channels. As for the CSPs, the flicker mutant E71Q behaved very differently and featured substantially and globally enhanced $R_{1rho}$ values with a maximum at V76. This means that the entire channel undergoes pronounced large-scale slow motions, which explains the strongly reduced sensitivity in our dipolar experiments with E71Q. Altogether, our data hence strongly suggest that the rapid flickering between open and closed states in KcsA, and observed in all K$^+$ channels[6,10], relates to large-scale dynamics of the pore domain.

Another noteworthy observation was the stark change in the dynamics of the turret, which rigidified in E71A and E71I. As we concluded above (Fig. 2c), this implies that the loss of the E71–D80 interaction causes allosteric changes 2 nm distal from the mutation site, presumably by successive changes of hydrogen bonding partners. A likely starting point for this chain reaction could be D80, which adopts a clearly different conformation in E71A and E71I compared to WT KcsA (Fig. 2a).

**Fluctuations in a critical water cavity trigger modal gating.** The strongly different filter dynamics in the E71X mutants raise the question which molecular events could cause spontaneous mode shifts. One potential trigger could be buried water, localised in a cavity behind the filter, which is a decisive gating determinant in K$^+$ channels, relating to C-type inactivation and recov-ery[22,31,32]. Previous X-ray studies suggested that changes in

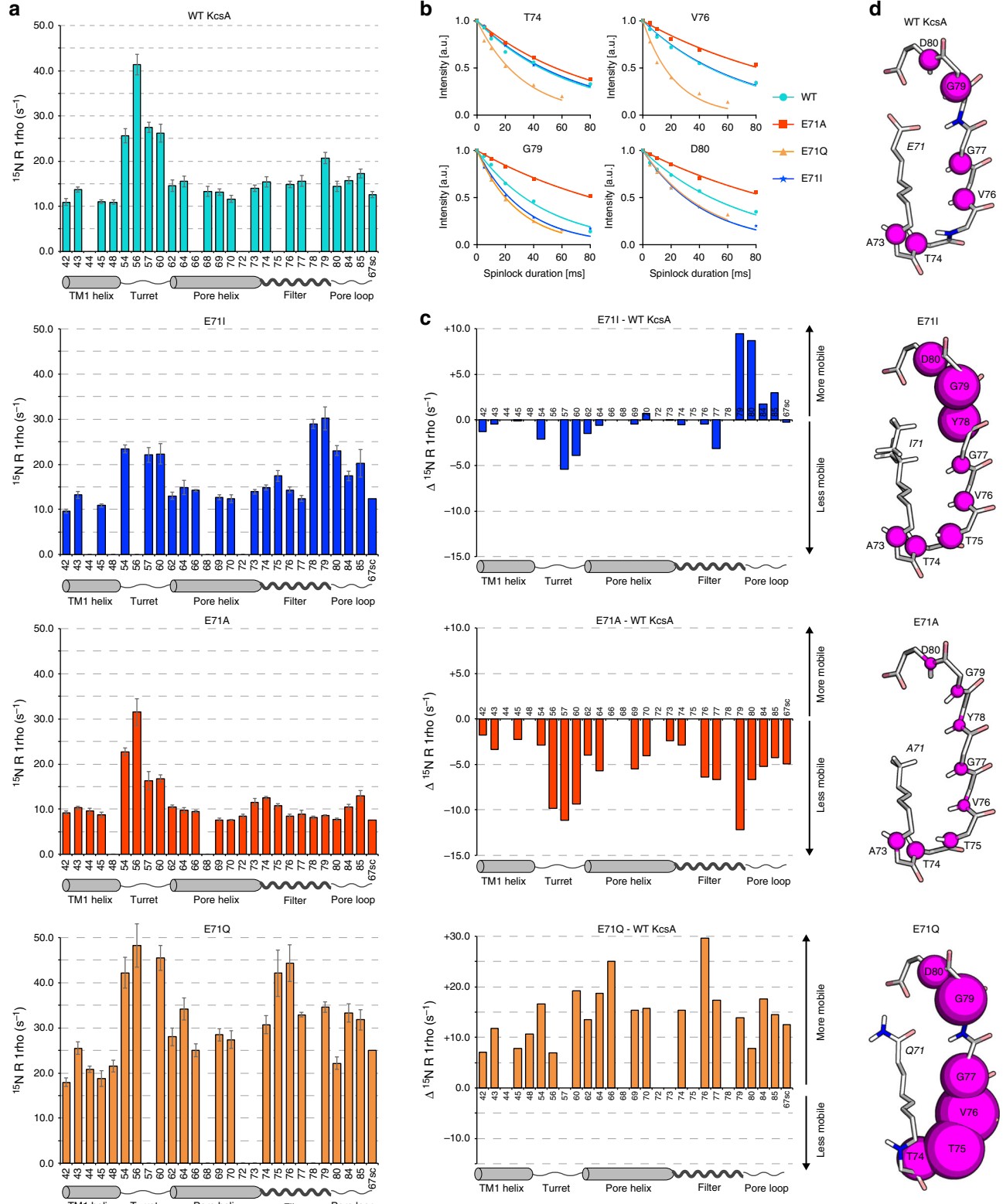

**Fig. 3** E71X point-mutations strongly change the selectivity filter dynamics. **a** $^{15}N$ rotating-frame ssNMR relaxation rates ($R_{1rho}$) that report on slow molecular motions in WT KcsA (cyan), E71A (red), E71I (blue), and E71Q (orange) measured at 700 MHz and 58 kHz MAS. The error bars show the standard error of the fit. Source data are provided as a Source Data file. **b** $R_{1rho}$ signal decay curves for selected filter residues. Symbols mark data points and lines represent best fits. **c** Plots of the differences in the dynamics between E71X mutants and WT KcsA. **d** Illustration of the site-resolved selectivity filter dynamics. The size of the magenta spheres represents the $R_{1rho}$ relaxation rates

the size of the water cavity could trigger modal gating shifts. In these and other previous studies, two water molecules were resolved behind the filter of E71I and E71A[6,12], while the presence of buried water molecules in E71Q was unclear due to insufficient resolution.

Here, we revisit the water distribution behind the selectivity filter using H/D exchange ssNMR[17,33,34], for which we acquired 2D $^{15}$N–$^1$H spectra in fully deuterated buffers. At these conditions, exchange with deuterons strongly attenuates signals of water-exposed amino-protons, which provides high-resolution information on the water cavity size[22]. The channels were incubated in deuterated buffers for 2 days, and the completeness of the exchange was confirmed on the fully water-exposed turret, which entirely disappeared from the 2D NH spectra (Fig. 4a, right panel). In WT KcsA, G79 is the only filter residue that disappeared in deuterated buffers while Y78–T74 showed no signs of H/D exchange (Fig. 4a, b). The lack of exchange for Y78 in WT KcsA is astonishing, given that a water molecule is in direct proximity in the X-ray structure (Fig. 1a), and strongly suggests that a tight hydrogen bond with E71 protects Y78 from H/D exchange. In line with this conclusion, Y78 showed attenuated intensity in both E71I (−33%) and E71A (−81%) (Fig. 4b, c), in which the X71–Y78 hydrogen bond is lost. Intriguingly, the much faster exchange of Y78 in E71A implies a larger water cavity, which agrees with the smaller size of alanine relative to isoleucine. The widened water cavity is also corroborated by the high rigidity of E71A (Fig. 3), which renders enhanced molecular fluctuations an unlikely cause for increased H/D exchange. Strikingly, in the flickery E71Q channel, G79 did not exchange (Fig. 4a). Similarly, L81 did not exchange in E71Q, while it disappeared in E71A, E71I, and WT KcsA, which confirms that the water cavity is smaller or fully absent behind the E71Q filter. Altogether, our NMR data demonstrate that, in membranes, E71X point-mutations change the size of the water cavity in reference to the gating mode, analogous to the crucial change of the water cavity during C-type inactivation[31]. Note that long MD simulations, which are discussed in detail in the following sections, also show widened water cavities and higher exchange-rates with bulk water for E71A and E71I (Supplementary Figure 4).

**Shifts in the equilibrium structure of the filter**. Our ssNMR data demonstrate that E71 point-mutations cause large CSPs in the selectivity filter, with maxima at Y78Cα and V76CO; the latter forming the S2 and S3 sites that are critically involved in C-type inactivation[3]. Such large perturbations in the heart of the filter are astonishing, given that E71A and E71I show a sharply reduced extent of C-type inactivation[6]. To gain a structural understanding of the ssNMR CSPs, we performed a series of 1-µs-long MD simulations for each WT KcsA and the mutants (Fig. 5a and Supplementary Figure 5). For WT KcsA, simulations show that the selectivity filter samples two conformations in which V76CO points either towards (inwards conformation) or away (outwards conformation) from the filter pore[35]. The equilibrium between these two, most likely rapidly converting, conformations was recently confirmed by 2D IR data that showed a 60:40 ratio between inwards:outwards conformations. Remarkably, simulations also indicate that E71X mutations perturb this equilibrium, stabilising the V76CO inwards conformation in E71I and E71A, while the outwards state is favoured in E71Q.

To probe if the V76CO CSPs could be explained by a change in the average conformation of the filter, we back-calculated[36] the $^{13}$C ssNMR chemical shifts over the MD simulations (Fig. 5b, c). These calculations yielded a much lower V76CO chemical shift in

the outwards conformation (predicted $\Delta$V76CO$_{inwards–outwards}$ = + 2 to + 3 ppm) (Fig. 5d); a result that agrees with the much lower chemical shift of V76CO at low [K$^+$] in the collapsed filter with a flipped V76–G77 plane (experimental $\Delta$V76CO$_{conductive\ filter–collapsed\ filter}$ = + 3.3 ppm)[21,37]. Hence, these results show that the V76CO inwards conformation is strongly stabilised in E71A (V76CO CSP = + 0.94 ppm relative to WT KcsA) and E71I (+0.77 ppm), while the sampling of the outwards conformation is enhanced in E71Q (−0.55 ppm) (Fig. 5e). This conclusion also agrees with our relaxation data, which show reduced dynamics for V76 and G77 in E71A and E71I, whereas both residues show much higher mobility in E71Q. Notably, the stabilisation of the V76CO inwards state correlates with the decrease of C-type inactivation in E71A and E71I[6].

Similarly, we could derive a structural understanding of the stark Y78Cα CSPs. In MD simulations, WT KcsA exclusively adopts the Y78CO inwards conformation that is stabilised by the E71–Y78 hydrogen bond. However, in E71I, the Y78 backbone is no longer stabilised and we observe sizeable sampling of an outwards conformation (Fig. 5a). Here again, back-calculated chemical shifts show a much lower Y78Cα signal in the Y78CO outwards conformation (predicted $\Delta$Y78Cα$_{inwards–outwards}$ = + 2 ppm) (Fig. 5c, d). These results imply that a Y78CO outwards state is frequently sampled in E71I (Y78Cα CSP = −1.94 ppm relative to WT KcsA) and less frequently or only partially in E71A (Y78Cα CSP = −0.84 ppm). Note that we could not observe a Y78CO outwards state in E71A simulations, presumably due to insufficient sampling.

**Destabilisation of the critical interaction D80–W67**. An especially remarkable finding in our study is the drastically increased filter dynamics in E71I compared to E71A despite similar nonpolar E71 substitutions, and this surge in flexibility most likely causes increased sampling of the Y78CO outwards state in E71I. We used MD simulations to gain a molecular understanding of the enhanced E71I filter dynamics. In our simulations, the conformational space of the D80 side chain is an event of particular interest (Supplementary Figure 6). In WT KcsA, the interaction with E71 locks D80 in a down conformation (Fig. 6a), and only this conformation enables the W67–D80 interaction, which is critical for gating in Kv channels[38,39]. The down conformation prevails in E71A, enabling a steady W67–D80 interaction (Fig. 6b, d). However, in E71I simulations, the bulky isoleucine sterically destabilises the W67–D80 interaction and causes D80 to increasingly sample middle and up conformations (Fig. 6c). We validated the loss of the W67–D80 interaction in E71I with ssNMR $^{15}$N $T_1$ relaxation experiments that are sensitive to the fast pico-to-nanosecond motion of unbound side chains. These experiments clearly show markedly enhanced W67 side chain dynamics in E71I, confirming an unstable W67–D80 interaction in this mutant (Fig. 6d, e). To compensate, D80 increasingly engages in interactions with the functionally important residues R64 and Y82[4,31] that are hardly sampled in WT KcsA. The D80 promiscuity is pronounced in MD simulations of E71I and E71Q and agrees with the enhanced ssNMR $R_{1rho}$ dynamics at the filter entrance of E71I and E71Q (Fig. 3). In E71A, however, a steady D80–W67 interaction stabilises the filter, as demonstrated by strongly reduced $R_{1rho}$ dynamics, and this stabilisation is most likely the reason why the Y78CO outwards state is much less frequently sampled in E71A than in E71I. Interestingly, our extensive MD analysis also demonstrates that the E71 point-mutations and the D80 promiscuity cause long-range modulations in the turret (Supplementary Figure 6), which presumably relate to the turret heterogeneity in the mutants (Figs. 2c and 3).

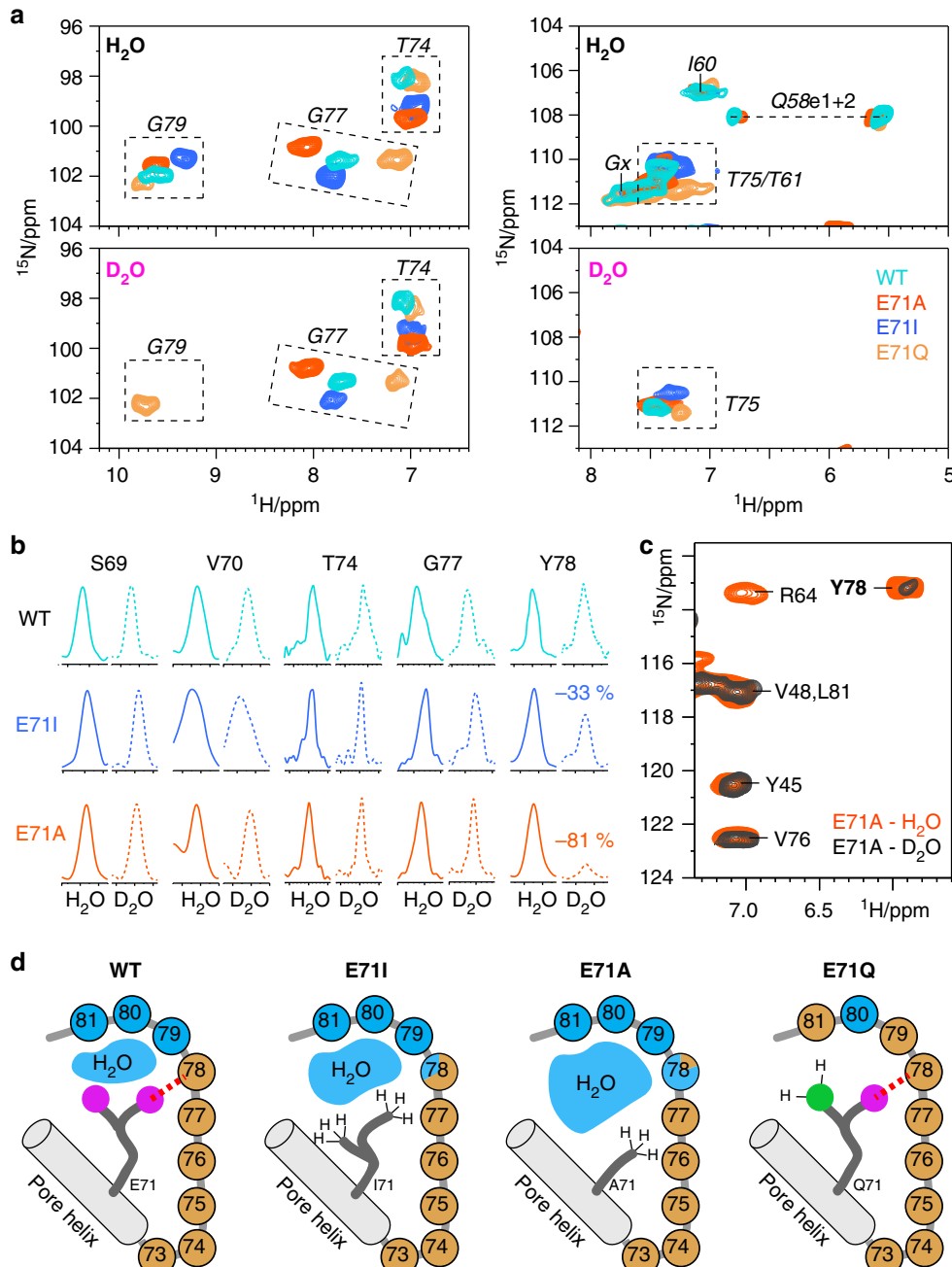

**Fig. 4** High-resolution analysis of the size of the water cavity behind the filter. **a** Zoom into 2D NH ssNMR spectra acquired in (upper panel) protonated and (lower) deuterated buffers of WT KcsA (cyan), E71I (blue), E71A (red), and E71Q (orange). **b** Cross-sections from 2D NH spectra of WT KcsA, E71I, and E71A measured in protonated (continuous lines) and deuterated (dashed) buffers. For Y78 in WT KcsA, cross-sections were extracted from 3D CANH experiments to resolve spectral overlap. Signals are normalised (see Methods). **c** 2D NH spectra of E71A in protonated (red) and deuterated (grey) buffers showing the fast exchange of Y78, implying a larger water cavity. **d** Illustrations of the ssNMR-derived water cavity size: in WT KcsA, the cavity is limited to G79-L81, and Y78 is exchange-protected. The cavity widens in E71I, strongly widens in E71A, and is absent in E71Q. Blue and brown spheres represent water-exposed and shielded amino-protons, respectively

## Discussion

Modal gating shifts at constant experimental conditions have been observed in K[+4,6–10], Na[+40], Ca[2+41,42], and other ion channels[43,44], and are considered a widespread regulatory mechanism[11,45], potentially to achieve intermediate activity levels. While known for a long-time, the molecular underpinning of modal gating behaviour is poorly understood. In KcsA, E71 point-mutations emulate modal gating shifts; however, X-ray structures of E71X mutants showed no differences compared to WT KcsA[6]. These seemingly disparate perspectives of functional heterogeneity and of structural similarity raise critical problems for our understanding of modal gating and also of Kv channel function (Fig. 1b). Our ssNMR study in native-like conditions paints a strikingly different picture, demonstrating that E71 substitutions lock the selectivity filter in characteristic conformational and motional landscapes that markedly diverge from WT KcsA. These landscapes strongly depend on the nature of residue 71 and directly relate to the heterogeneous functional behaviour observed in K[+] channels[6].

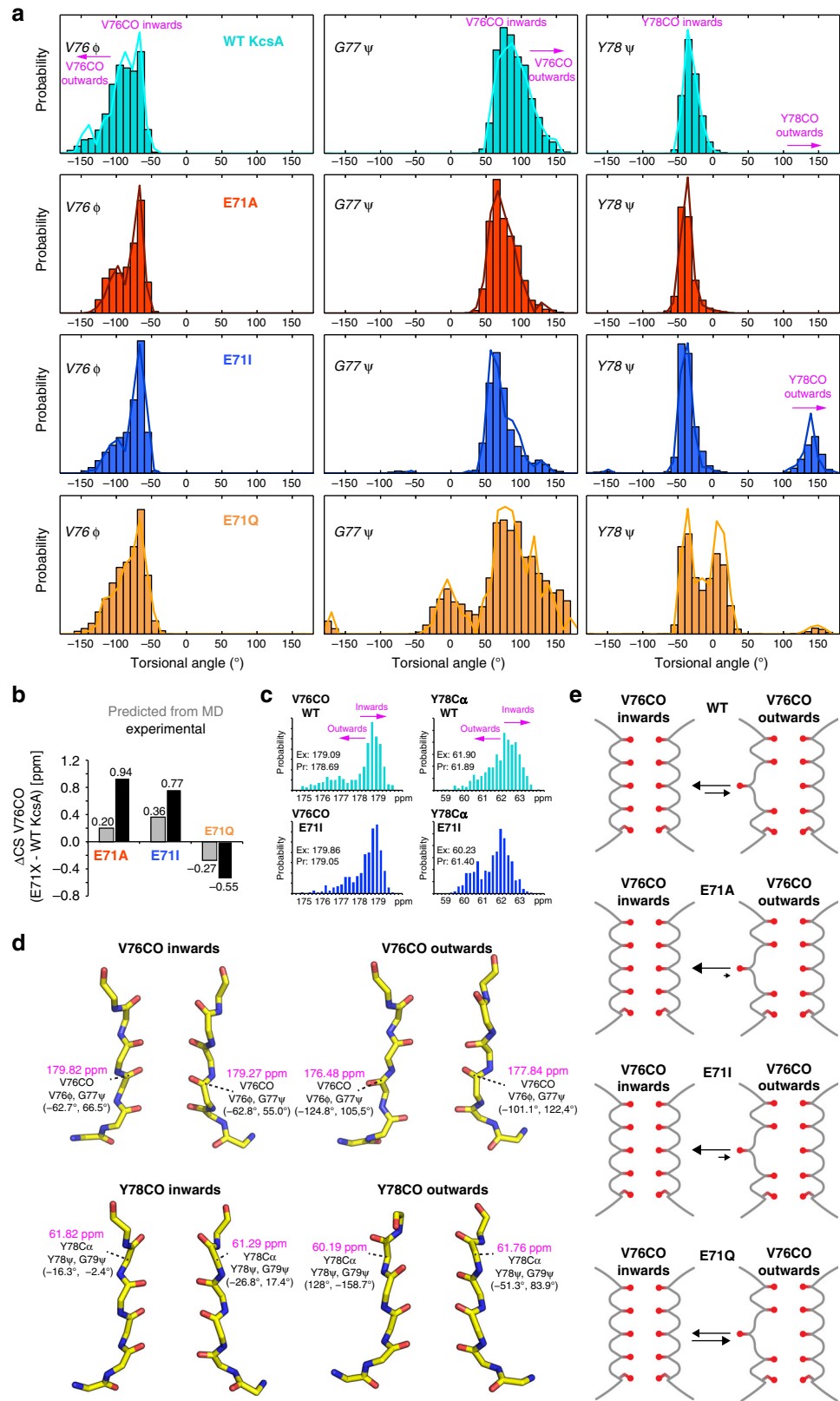

By integrating ssNMR and MD simulations, we show that E71 point-mutations rearrange the network behind the filter and perturb the K⁺ binding sites V76 and Y78 (Fig. 1a, e). Thereby, we show that E71X mutations change the equilibrium between intrinsically sampled filter states (Fig. 5e), which agrees with 2D IR data[35] and the so-called 'flipped' E71A structure that points towards a complex selectivity filter landscape that includes dynamical flips of K⁺ coordinating peptide planes[35]. Our data demonstrate a stabilisation of the V76CO inwards state in E71A and E71I relative to WT KcsA, which correlates with a sharply reduced entry into the C-type inactivated state. Furthermore, our data clearly show Y78 conformational perturbations in E71I and

**Fig. 5** E71X mutations shift the equilibrium between inwards and outwards filter states. **a** Dihedral angle distribution of filter residues of WT (cyan), E71A (red), E71I (blue), and E71Q (orange) derived from 1-μs-long MD simulations. Characteristic angular spaces for inwards conformations, with the carbonyl group oriented towards the filter pore, and outwards states are highlighted. **b** Comparison of V76CO CSPs derived from experiments (black bars) and back-calculated[36] from MD simulations (grey bars). Source data are provided as a Source Data file. **c** Histogram of back-calculated chemical shifts for V76CO and Y78Cα of WT KcsA (cyan) and E71I (blue). The V76CO (left) inwards state is stabilised in E71I, leading to higher V76CO chemical shifts, while the Y78CO (right) inwards state is destabilised in E71I, leading to lower Y78Cα chemical shifts. **d** Representative MD snapshots of WT KcsA and E71I showing inwards and outwards states of V76CO and Y78CO. The chemical shifts (in magenta) of V76CO and Y78Cα strongly differ between inwards and outwards conformations. **e** Illustration of the stabilisation of the V76CO inwards state in E71A and E71I, and the destabilisation in E71Q

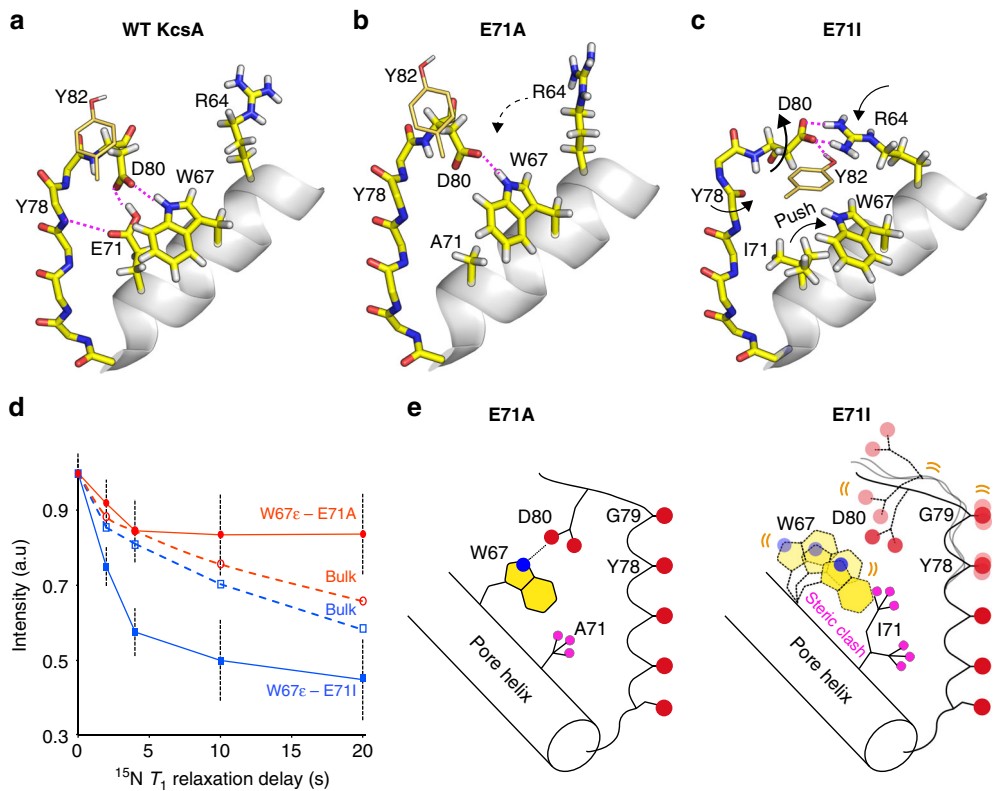

**Fig. 6** The functionally critical W67–D80 interaction is destabilised in E71I KcsA. **a** WT KcsA MD simulation: the tight interaction with E71 locks the D80 side chain in a down configuration that enables hydrogen bonding with W67 (snapshot after 270 ns). **b** E71A simulation: the down conformation prevails, enabling the W67–D80 interaction, which stabilises the filter entrance (snapshot after 600 ns). **c** E71I simulation: I71 impedes the W67–D80 interaction which destabilises the filter entrance. D80 engages in interactions with Y82 (from a neighbouring channel subunit) and R64 (snapshot after 600 ns). **d** Longitudinal relaxation times ($^{15}$N $T_1$) that report on fast motion of the W67 side chain for E71A (red circles) and E71I (blue squares), measured at 950 MHz and 60 kHz MAS. The error shows the signal-to-noise ratio for W67Nε at a given data point. Source data are provided as a Source Data file. **e** (left) The W67–D80 interaction is maintained in E71A; (right) I71 hinders the W67–D80 hydrogen bond, which entails increased dynamics at the pore mouth

E71A. Such perturbations could not be observed in WT KcsA X-ray structures[3]; however, there is strong evidence from previous ssNMR studies that Y78 modulations can accompany filter gating. Indeed, ssNMR data show that Y78Cα exhibits a drastically lower signal in the open-collapsed state (experimental ΔY78Cα$_{\text{conductive filter–collapsed filter}}$ = +4.3 ppm)[37], which unambiguously argues that the Y78 conformation can change in reference to the filter mode. This notion is also corroborated by the 'flipped' E71A X-ray structure (2ATK), which also features a Y78CO partial outward state[4,46], and is in line with a recent cryo-EM structure of the hERG channel[47]. While the exact role in KcsA is unclear, we surmise that Y78 backbone modulations may relate to a non-conductive state that is favoured by E71I, which agree with the strongly reduced K$^+$ occupancy at S0–S2 in E71I[6].

Importantly, the marked conformational changes in the E71I selectivity filter, and especially the destabilisation of the W67–D80 interaction (Fig. 6d, e), suggest to be of broad relevance for Kv channels (Fig. 1b) where the W67–D80 interaction plays a defining role in the inactivation process. Mutational studies showed that the inability to establish this highly conserved interaction entails severe functional perturbations for KcsA, Shaker, and Kv1.2[38,39]. Moreover, in line with the effect of the weakening of the W67–D80 interaction in E71I, the destabilisation of the analogous interaction W434–E447 in Shaker modulates the equilibrium between conducting and non-conducting filter states[48].

On the basis of our set of results, we show that modal gating shifts in K$^+$ channels relate to changes in the statistical weighting of pre-existing selectivity filter states which are triggered by

fluctuations in the hydrogen bonding (Fig. 6) and water network (Fig. 4). Notably, we show that modulations in this network cause changes in the turret over more than 2 nm (Figs. 2c, 3), which opens a pathway how turret-binding drugs, lipids, or other proteins can allosterically modulate the filter[21,25,49–51]. The question arises why most of these conformational subtypes could not be crystallised. Here, the reason is most likely the interaction with Fab fragments that act as crystallographic chaperons and attach to KcsA X-ray structures. In agreement with electrophysiological measurements[4], these artificial Fab interactions lock the selectivity filter in a specific conformation, and thereby hinder the capturing of transient configurations, masking the effect of E71 point-mutations. Interestingly, the lack of non-canonical filter conformations in X-ray structures was mirrored by additional torsional or position restraints in MD simulations to stabilise a conductive filter conformation[52,53]. At least for KcsA, such potentials could mask the physiological filter plasticity, as demonstrated in this study and previously with 2D IR[35].

In conclusion, our work establishes the shifts in the conformational dynamics of the selectivity filter as the key physiological determinant of modal gating behaviour. At the same time, our work provides a long-needed quantitative description of the selectivity filter dynamics in a native environment, which is of fundamental importance to understand ion channel function[54]. Given that the here-described filter dynamics are strongly different in mutant E71I, our study may ultimately help to better understand eukaryotic Kv channels. Finally, we like to emphasise that further experiments with open channels under inactivating conditions will be critically required to fully comprehend how E71X mutations modulate channel open probability and C-type inactivation.

## Methods

**Sample preparation**. WT KcsA and E71X mutant channels were expressed in *E. coli* M15 cells (Qiagen) using standard $H_2O$-based M9 medium supplemented with 0.5 g/L $^{15}NH_4Cl$ and 2 g/L D-glucose-$^{13}C6$-d7 in order to improve the spectral resolution in $^1H$-detected ssNMR experiments[17]. Cells were subsequently harvested, treated with lysozyme, and lysed via French press. The membranes containing the KcsA channels were collected by centrifugation (100,000×$g$) and proteins were extracted with 40 mM DM (Anatrace)[26]. KcsA channels were purified using Ni-NTA agarose beads (Qiagen), resulting in a final yield of 10 mg/L for WT KcsA and 5 mg/L for the E71X KcsA mutants. Liposome reconstitution was performed using *E. coli* polar lipids (Avanti) at 1:100 protein:lipid molar ratio, in which the detergent was removed using polystyrene beads (Bio-Beads SM-2)[26]. Before the ssNMR measurements, reconstituted samples were suspended in fully protonated phosphate buffer (pH 7.4, 100 mM $K^+$). For spectral assignments, a fully protonated buffer was used in order to observe the entire channel in $^1H$-detected ssNMR experiments. For proton/deuterium (H/D) exchange ssNMR spectroscopy, ion channels were incubated in fully deuterated buffers (pH 7.4, 100 mM $K^+$) for a total of 2 days prior to the measurements.

**Solid-state NMR spectroscopy**. 3D ssNMR experiments for sequential backbone chemical shift assignments were performed at 800 MHz ($^1H$-frequency) using 60 kHz magic angle spinning (MAS) frequency and a real temperature of approximately 305 K. In total, we ran ten dipolar-based 3D ssNMR experiments to assign the three mutant channels and WT KcsA. The pulse sequences and experimental setups were performed as previously described[17]. 2D $^{13}C$–$^{13}C$ PARISxy[55] ($N = \frac{1}{2}$, $m = 1$) experiments for side chain chemical shift assignments were performed at 700 MHz using 42 kHz MAS and a $^{13}C$–$^{13}C$ mixing time of 110 ms. $^{15}N$ $T_{1rho}$ relaxation experiments were performed as described for the water-inaccessible part of KcsA and measured at 700 MHz and 58 kHz MAS using a $^{15}N$ spin lock amplitude of 17.5 kHz[17]. We used $^1H$-detected 2D experiments together with relaxation increments of 0, 5, 10, 20, 40, and 80 ms. For the much faster relaxing flicker E71Q channel, we used increments of 0, 5, 10, 20, 40, and 60 ms. $^{15}N$ $T1$ measurements were performed at 950 MHz and 60 kHz MAS using relaxation elements of 0, 2, 4, 10, and 20 s. The W67 side chain is spectrally isolated in E71A and E71I (Supplementary Figure 2) and could be readily analysed in a series of 1D experiments. For the analysis of H/D exchange data from 2D NH spectra acquired in protonated and deuterated buffers, we used the signal intensities, which were normalised to the water-inaccessible residues S69 and V70 that are not subjected to H/D exchange. PISSARRO[56] decoupling was used as

decoupling method in all direct and indirect dimensions. Chemical shifts were back-calculated from MD simulations with the SPARTA+ program[36]. Channels structures were extracted from MD simulations with a time-increment of 10 ns, yielding a total of 400 monomer structures. Compared to our experimental data, the SPARTA+ predictions give systematically lower chemical shifts for the V76 backbone carbonyl carbon, which we aligned by adding 2.5 ppm to the predicted carbonyl chemical shifts for all channels.

**MD simulations**. Atomic models of KcsA were constructed based on crystal structures (1K4C[3] and 3OR6[6]) that represent a structural state with closed inner gate. Considering the high similarity between crystal structures of WT (1K4C), E71A (1ZWI), and E71I (3OR7), the simulation systems for E71A and E71I are built based on the fully equilibrated WT system by introducing respective single mutation. With more substantial difference compared with WT, the crystal structure for E71Q (3OR6) was used to build its simulation system. For all MD simulations, the channel was embedded in a bilayer of 3:1 POPC:POPG lipids and solvated in 150 mM or 200 mM KCl using the web service CHARMM-GUI[57,58]. Most residues were assigned their standard protonation state at pH 7. The total number of atoms in the MD systems is on the order of ~49,000. The CHARMM force field PARAM36 for protein[59], lipids[60], and ions[61] was used. Explicit water was described with the TIP3P model. In WT, the residue Glu71 is protonated to form a key hydrogen bond with Asp80[39,62]. The models of KcsA were refined using energy minimization for at least 2000 steps, and the ions and non-filter backbone atoms were kept fixed throughout the minimization procedure. After energy minimisation, the conductive filter was restrained for 10–20 ns to relax any unfavourable contacts destabilising the filter. All the simulations were performed under constant NPT conditions at 310 K and 1 atmosphere, and periodic boundary conditions with electrostatic interactions were treated by the particle-mesh Ewald (PME) method and a real-space cutoff of 12 Å. The simulations use a time step of 2 fs. After minimization and equilibration with harmonic positional restraints on all of the C atoms, MD simulations were performed for 1 μs for wild type and all mutants, by using either NAMD version 2.11[63], or on the special purpose computer Anton (Pittsburgh Supercomputer Center)[64].

**Reporting Summary**. Further information on experimental design is available in the Nature Research Reporting Summary linked to this article.

## Data availability
Data supporting the findings of this manuscript are available from the corresponding author upon reasonable request. The solid-state NMR assignments have been deposited in the BMRB (accession number 27676 for WT KcsA, 27678 for E71A KcsA, 27679 for E71I KcsA, and 27680 for E71Q KcsA). The Source Data underlying Figs. 1d, 3a, 5b, 6d, and Supplementary Figure 3 are provided as a Source Data file.

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

## Acknowledgements

This work was funded in part by the National Institutes of Health/National Institute of General Medical Sciences (NIH/NIGMS) through grants R01-GM062342 and U54-GM087519, and the Netherlands Science Organisation for Scientific Research (NWO, grant numbers 723.014.003 & 711.018.001 to M.W.; 700.26.121 to M.B.). Experiments at the 950 MHz instrument were supported by uNMR-NL, an NWO-funded Roadmap NMR Facility (no. 184.032.207).

## Author contributions

S.J., J.M.S., and F.K. acquired the ssNMR data. J.L. and B.R. performed the MD simulations. All authors contributed to data analysis and drafting of the manuscript. All authors critically reviewed the paper.

## Additional information

**Competing interests:** The authors declare no competing interests.

