## [Peer Review File · Nature Communications]

Reviewers' Comments:

Reviewer #1:

Remarks to the Author:

This work from Jekhmane et al investigates the molecular basis for modal gating shifts in KcsA using solid state NMR and molecular dynamic simulations. Although modal gating is a common phenomenon in ion channels, it is not clear what causes it. It was previously reported that mutations at a site on the pore helix alters gating kinetics in a side-chain specific way. However, this study did not provide conclusive mechanistic insights. Here, the authors study the dynamics of the selectivity filter (SF) region of WT and E71X mutants in reconstituted liposomes. By combining ssNMR data with MD simulations they elucidate the role of the hydrogen-bond network and water molecules behind the SF in the stabilization of the gating modes. In addition, their study reveals several previously unobserved gating conformational changes in the turret region of the SF which was masked in the crystal structures, potentially due to the antibody binding. Overall, this an exciting work and the manuscript is very well-written. I only have a few minor comments.

1. The authors need to mention that the data presented here are on the closed channel (at neutral pH) and the observed modal shifts are a function of the open channel (acidic pH). The underlying assumption being made here needs to be explicitly stated.
2. It would be useful to show, perhaps as a cartoon representation, what the authors mean by high-open probability, low-open probability, and high-frequency flicker modes.
3. What is a "closed-conductive" channel? (first paragraph of results section). I understand what the authors mean, but it may not be obvious to the readers.

Reviewer #2:

Remarks to the Author:

The paper by Weingarth and co-workers addresses the molecular foundations of 'modal gating shifts' in the ion channel KcsA. This phenomenon has been observed in most ion channels and has been recognized as a widespread regulator mechanism. The structural foundations for these different modes and how they switch, has not been resolved yet.

The approach presented here for KcsA is based on the analysis of changes in structure and dynamics of the filter region various KcsA E71X single point mutants. Each of these mutants corresponds to specific gating modes. Solid-state NMR has been used as a spectroscopic readout since it is directly applicable to KcsA embedded within lipid bilayers. The authors present a series of 2D ^{13}C - ^{13}C and ^{15}N - ^1H spectra to obtain mutation-induced chemical shift changes, determined ^{15}N -R1rho relaxation rates, which are linked to microsecond dynamics and also probed H/D exchange around the selectivity filter. Clear differences between the mutants have been obtained, which can be linked to the different gating modes, which is also supported by MD simulations.

Overall, the data are of very good quality obtained by advanced spectroscopy and the scientific story is well presented. Most likely, the obtained data will have a great impact towards understanding the activity of ion channels.

The only question I have relates to important inter-residue interactions such as between D80-W67 (Fig. 6E) or E71-D80 (as in Fig. 2a). I could not find any data in the paper demonstrating directly their interactions or mutation-induced loss of these contacts. Could the authors comment on that?

Reviewer #3:

Remarks to the Author:

This manuscript uses a combination of NMR spectroscopy and MD simulations to explore conformational changes in the selectivity filter of the K^+ channel KcsA. They show that mutations that shift the selectivity filter between PO-high, PO-low, and flickering states also impact the dynamics and conformation of the selectivity filter. Dynamics in the filter are important

considerations, and likely very related to C-type inactivation, but solid experimental evidence supporting these conformational changes have been really lacking. To this extent, this manuscript does a very nice job of exploring these changes. This paper is important to the field.

I have a few questions that I would like to be addressed:

1. I got confused by the statement on page 8:

"Notably, the stabilisation of the V76CO inwards state correlates with the decrease of C-type inactivation in E71A and E71I."

In particular, Figure 6C made me think – why isn't E71I going into C-type inactivation mode more? It wasn't until I went back and reread the Chakrapani paper in NSMB (2011) that I really remembered that while these mutants all have a propensity for different PO-high, PO-low, and flickering states, they all adopt the full closed state less (nicely shown in Fig. 1 from that paper). I am sure you said that in the paper, but perhaps make it clearer, and perhaps recap these mutants a little better?

2. I do not follow the argument about the size of the water cavity based on the H/D exchange (Fig. 4, page 7). Isn't it possible that the cavity could still be small in E71A, but the fluctuations in the strand are bigger, and hence we get exchange? The other idea is that the hydronium is penetrating into the water filled cavity behind the filter even without strand fluctuations, and the exchange is happening from there, and I think that readers would appreciate this too.

3. Related to point #2 above, what do the simulations show for the size of the water cavities in these mutants/WT, and the exchange rate with waters from solution?

Minor points

1. Page 5. Sentence starting, "Experimental data if changes in the filter ..." rewrite this, it is hard to follow.
2. Page 6. Sentence starting, "Intriguingly, the motional selectivity filter landscape ..." This is an awkward phrase.
3. Page 8. Sentence starting, "An especially remarkable finding in our study is the drastically filter dynamics ..." grammar.
4. Page 11. Sentence starting, "Importantly, the here revealed .." grammar.

Response to the Reviewers' Comments on the Manuscript NCOMMS-18-30366-T entitled " Shifts in the selectivity filter dynamics cause modal gating in K⁺ channels" submitted to Nature Communications

We would like to thank the reviewers for carefully reading our paper and for their critique. We are very encouraged by the very positive evaluation of our manuscript by all three reviewers. The valuable comments from all reviewers have allowed us to improve our manuscript. In the following, we include a point-by-point response to the questions and comments of each reviewer.

Reviewer #1:

We are very grateful for her/his very positive opinion on our manuscript.

Minor points:

(1) The authors need to mention that the data presented here are on the closed channel (at neutral pH) and the observed modal shifts are a function of the open channel (acidic pH). The underlying assumption being made here needs to be explicitly stated.

We thank the reviewer for her/his attentiveness, and we fully agree with her/his comment. Therefore, we added an explicit passage in the discussion that our understanding of gating cycle in the different mutants is yet compromised and requires detailed studies of the inactivated states. We have clarified this on page 12:

"Finally, we like to emphasize that our molecular understanding of the impact of E71X mutations on the gating cycle of K⁺ channels is yet compromised. Here, experiments with open channels under inactivating conditions will be critically required to fully comprehend how E71X mutations modulate channel open probability and C-type inactivation."

(2) It would be useful to show, perhaps as a cartoon representation, what the authors mean by high-open probability, low-open probability, and high-frequency flicker modes.

We thank the reviewer for her/his comment. We agree with the referee that illustrative single channel traces for the different mutants are helpful to introduce modal gating to the reader. However, given that detailed, representative single channel traces for the mutants have been published (Perozo & Roux, NSMB 2011), we would actually prefer referring the reader to the original works that are cited in our manuscript. We are also unsure that a cartoon representation could live up to the complexity of the subject.

(3) What is a “closed-conductive” channel? (first paragraph of results section). I understand what the authors mean, but it may not be obvious to the readers.

We thank the reviewer for her/his criticism. For clarification we added on page 2:

..at which WT KcsA is in the closed-conductive state, i.e., a state with a closed activation gate and a conductive selectivity filter.

Reviewer #2:

We thank the reviewer very much for her/his very positive evaluation of our manuscript. We are especially encouraged by her/his comment that our work 'will have a great impact towards understanding the activity of ion channels'

Minor point:

(1) The only question I have relates to important inter-residue interactions such as between D80-W67 (Fig. 6E) or E71-D80 (as in Fig. 2a). I could not find any data in the paper demonstrating directly their interactions or mutation-induced loss of these contacts. Could the authors comment on that?

We thank the reviewer very much for her/his comment. We did not probe interactions D80 – W67 or E71 – D80 in WT KcsA at pH7 because a large body of evidence (functional studies, crystal structures, computational studies) has confirmed these interactions over the last 20 years.

Regarding the mutants, the referee is fully correct that we do not probe these interactions by dipolar NMR contacts. We could not engage in such dipolar NMR studies because, very unfortunately, none of the A71, I71, or Q71 sidechains are resolved in 2D CC spectra. However, we like to emphasize that we have nevertheless abundant and compelling information on the X71 – D80 and D80 – W67 interactions in the mutants derived from

- Chemical shift perturbations (Figure 1D and 2A)
- H/D exchange data (Figure 4B)
- ¹⁵N T₁ relaxation data (Figure 6D,E)
- Long MD simulations (Figure 6A-C and Supporting Figures 5 and 6)

Hence, while we fully agree with the referee that direct dipolar contacts between X71 – D80 and D80 – W67 would be ideal, the total of our insights provides us with coherent and detailed information on these interactions in WT KcsA and the mutant channels.

Reviewer #3:

We are very grateful for her/his very positive opinion on our manuscript.

Major points:

(1) I got confused by the statement on page 8:

“Notably, the stabilisation of the V76CO inwards state correlates with the decrease of C-type inactivation in E71A and E71I.”

In particular, Figure 6C made me think – why isn't E71I going into C-type inactivation mode more? It wasn't until I went back and reread the Chakrapani paper in NSMB (2011) that I really remembered that while these mutants all have a propensity for different PO-high, PO-low, and flickering states, they all adopt the full closed state less (nicely shown in Fig. 1 from that paper). I am sure you said that in the paper, but perhaps make it clearer, and perhaps recap these mutants a little better?

We thank the reviewer very much for her/his comment. We like to emphasize that our manuscript deals with closed channels, while the open probabilities in the Chakrapani paper (NSMB) are a function of open channels. To fully understand the nature of the “slow” inactive state, intermediate inactive state, and flicker state, we need to perform systematic studies for these mutations in inactive condition. To clarify this point, we have added in the Discussion section on page 12:

“Finally, we like to emphasize that our molecular understanding of the impact of E71X mutations on the gating cycle of K⁺ channels is yet compromised. Here, experiments with open channels under inactivating conditions will be critically required to fully comprehend how E71X mutations modulate channel open probability and C-type inactivation.”

(2) I do not follow the argument about the size of the water cavity based on the H/D exchange (Fig. 4, page 7). Isn't it possible that the cavity could still be small in E71A, but the fluctuations in the strand are bigger, and hence we get exchange? The other idea is that the hydronium is penetrating into the water filled cavity behind the filter even without strand fluctuations, and the exchange is happening from there, and I think that readers would appreciate this too.

We thank the reviewer very much for her/his critique. We agree with the reviewer that, in principle, strongly enhanced molecular fluctuations could also relate to an augmented water-exchange in the back of the E71A filter. However, our site-resolved relaxation data (Figure 3) show that the E71A filter and pore loop are extraordinarily stiff and hence rule out that enhanced dynamics are responsible for increased H/D exchange in E71A. We clarified this on page 8 of the manuscript:

“The widened water cavity is also corroborated by the high rigidity of E71A (Figure 3), which renders enhanced molecular fluctuations an unlikely cause for increased H/D exchange.”

Regarding the penetration of a hydronium ion in the absence of fluctuations: unfortunately, we are not aware of experimental data for the presence of hydronium ions in or around the K^+ channel pore. In our MD simulations, in which we use neutral water molecules, we did not observe penetration from the pore towards the cavity behind the filter.

(3) Related to point #2 above, what do the simulations show for the size of the water cavities in these mutants/WT, and the exchange rate with waters from solution?

We are grateful for the referee’s comment. Our MD simulations also show widened water cavities for E71I and especially E71A, in agreement with our experimental NMR data.

Figure. 2D average occupancy map for all four subunits during 1000ns MD simulations. The x-axis describes the radius to the center of the selectivity filter, and the y-axis is the z-coordinate of water molecules.

Regarding the exchange rates, we observe that water molecules have a higher turnover rate in E71I and E71A relative to WT KcsA, which is also in line with an increased H/D exchange and larger water cavities in these mutants.

MD derived water exchange rates with bulk water

	WT	E71I	E71A	E71Q
# of water molecules	0.81±0.12	1.47±0.21	2.30±0.16	1.58±0.16
Turnover time (ns)	51.9±40.9	7.51±4.82	18.5±8.9	6.7±1.9

We discuss the computationally derived water distribution and exchange rates in detail in the Supplementary Information (Supplementary Figure 4), where we also discuss potential reasons and present ssNMR data for the increased exchange with bulk water in the mutant channels. In Supplementary Figure 4, we also discuss potential reasons why the MD simulations do not reproduce our NMR data for the water occupancy in E71Q.

Furthermore, we added this new computational analysis to the main text (page 8):

“Note that long MD simulations, which are discussed in detail in the following sections, also show widened water cavities and higher exchange rates with bulk water for E71A and E71I (Supplementary Figure 4).”

Minor points:

We thank the referee very much for improving the style and grammar of our manuscript.

(4) Page 5. Sentence starting, “Experimental data if changes in the filter ...” rewrite this, it is hard to follow.

We rephrased this to:

“quantitative experimental data on selectivity filter dynamics are not available in membranes”

(5) Page 6. Sentence starting, “Intriguingly, the motional selectivity filter landscape ...” This is an awkward phrase.

We rephrased this to:

“Intriguingly, the selectivity filter dynamics”

(6) Page 8. Sentence starting, “An especially remarkable finding in our study is the drastically filter dynamics ...” grammar.

A word was missing:

“An especially remarkable finding in our study is the drastically increased filter dynamics ...”

(7) Page 11. Sentence starting, “Importantly, the here revealed ..” grammar.

We rephrased this to:

Importantly, the here-revealed marked conformational changes...”

Reviewers' Comments:

Reviewer #2:

Remarks to the Author:

The authors have fully addressed all of my remaining concerns.

Reviewer #3:

Remarks to the Author:

I am happy with the changes/clarifications shown by the authors. I would suggest that they tone down their use of the phrase "... is yet compromised." In the final paragraph. This is far too strong. Simply state that additional experiments must be performed to probe the filter dynamics of channels with open activation gates.

Response to the Reviewers, 2nd revision

We would like to thank the reviewers very much for carefully reading our manuscript and for their critique. In the following, we include a response to the remaining comment of reviewer 3.

Reviewer #3:

Minor point:

I am happy with the changes/clarifications shown by the authors. I would suggest that they tone down their use of the phrase "... is yet compromised." In the final paragraph. This is far too strong. Simply state that additional experiments must be performed to probe the filter dynamics of channels with open activation gates.

We are very grateful for her/his feedback, which has significantly improved the quality of our manuscript. Accordingly, we changed on page 10:

"Finally, we like to emphasize that our molecular understanding of the impact of E71X mutations on the gating cycle of K⁺ channels is yet compromised. Here, experiments with open channels under inactivating conditions will be critically required to fully comprehend how E71X mutations modulate channel open probability and C-type inactivation."

to

"Finally, we like to emphasize that ~~further our molecular understanding of the impact of E71X mutations on the gating cycle of K⁺ channels is yet compromised. Here,~~ experiments with open channels under inactivating conditions will be critically required to fully comprehend how E71X mutations modulate channel open probability and C-type inactivation."